# Compositional De-Attention Networks

[†]**Yi Tay**,[*] [♯]**Luu Anh Tuan**[*], [♮]**Aston Zhang**, [♣]**Shuohang Wang**, [♭]**Siu Cheung Hui**
[†,♭]Nanyang Technological University, Singapore
[♯]MIT CSAIL, [♮]Amazon AI
[♣]Microsoft Dynamics 365 AI Research
ytay017@gmail.com

## Abstract

Attentional models are distinctly characterized by their ability to learn relative importance, i.e., assigning a different weight to input values. This paper proposes a new quasi-attention that is compositional in nature, i.e., learning whether to *add*, *subtract* or *nullify* a certain vector when learning representations. This is strongly contrasted with vanilla attention, which simply re-weights input tokens. Our proposed *Compositional De-Attention* (CoDA) is fundamentally built upon the intuition of both similarity and dissimilarity (negative affinity) when computing affinity scores, benefiting from a greater extent of expressiveness. We evaluate CoDA on six NLP tasks, i.e. open domain question answering, retrieval/ranking, natural language inference, machine translation, sentiment analysis and text2code generation. We obtain promising experimental results, achieving state-of-the-art performance on several tasks/datasets.

## 1 Introduction

*Not all inputs are created equal*. This highly intuitive motivator, commonly referred to as *'attention'*, forms the bedrock of many recent and successful advances in deep learning research [Bahdanau et al., 2014, Parikh et al., 2016, Seo et al., 2016, Vaswani et al., 2017]. To this end, the Softmax operator lives at it's heart, signifying the importance of learning relative importance as a highly effective inductive bias for many problem domains and model architectures.

This paper proposes a new general purpose quasi-attention method. Our method is *'quasi'* in the sense that it behaves like an attention mechanism, albeit with several key fundamental differences. Firstly, instead of learning relative importance (weighted sum), we learn a compositional pooling of tokens, deciding whether to *add*, *subtract* or *delete* an input token. Since our method learns to flip/subtract tokens, deviating from the original motivation of attention, we refer to our method as a quasi-attention method. Secondly, we introduce a secondary de-attention (deleted attention) matrix, finally learning a multiplicative composition of similarity and dissimilarity. We hypothesize that more flexible design can lead to more expressive and powerful models which will arrive at better performance.

In order to achieve this, we introduce two technical contributions. The first, is a dual affinity scheme, which introduces a secondary affinity matrix $N$, in addition to the original affinity matrix $E$. The affinity matrix $E$, commonly found in pairwise [Parikh et al., 2016] or self-attentional [Vaswani et al., 2017] models, learns pairwise similarity computation between all elements in a sequence (or two sequences), i.e., $e_{ij} = a_i^\top b_j$. Contrary to $E$, our new $N$ matrix is learned a dissimilarity metric such as negative $L1$ distance, providing dual flavours of pairwise composition.

---

[*]Denotes equal contribution.

Secondly, we introduce a compositional mechanism which composes $tanh(E)$ with $sigmoid(N)$ to form the quasi-attention matrix $M$. In this case, the first term $tanh(E)$ controls the adding and subtracting of vectors while the secondary affinity $N$ can be interpreted as a type of gating mechanism, erasing unnecessary pairwise scores to zero when desired. The motivation for using dissimilarity as a gate is natural, serving as a protection against over-relying on raw similarity, i.e, if dissimilarity is high then the negative component learns to erase the positive affinity score.

Finally, the quasi-attention matrix is then utilized as per standard vanilla attention models, pooling across a sequence of vectors for learning attentional representations. More concretely, this new quasi-attention mechanism is given the ability to express arithmetic operations [Trask et al., 2018] when composing vectors, i.e., compositional pooling. As a general-purpose and universal neural component, our CoDA mechanism can be readily applied to many state-of-the-art neural models such as models that use pairwise attention or self-attention-based transformers.

All in all, the prime contributions of this work are as follows:

- We introduce *Compositional De-Attention* (CoDA), a form of quasi-attention method. Our CoDA mechanism is largely based on two new concepts, (1) dual affinity matrices and (2) compositional pooling, distinguishing itself from all other attention mechanisms in the literature.

- Our CoDA method decouples the Softmax operator with standard attention mechanisms and puts forward a new paradigm for attentional pooling in neural architectures. To the best of our knowledge, this is the first work that explores the usage of Softmax-less attention mechanisms. As a by-product, we also show that going Softmax-less could also be a viable choice even in attentional models.

- CoDA enables a greater extent of flexibility in composing vectors during the attentional pooling process. We imbue our model with the ability to subtract vectors (not only relatively weight them).

- We conduct extensive experiments on a myriad of NLP tasks such as open domain question answering, ranking, natural language inference, machine translation, sentiment analysis and text2code generation. We obtain reasonably promising results, demonstrating the utility of the proposed CoDA mechanism, outperforming vanilla attention more often than not. Moreover, CoDA achieves state-of-the-art on several tasks/datasets.

## 2 Compositional De-Attention Networks (CoDA)

This section introduces the proposed CoDA method.

### 2.1 Input Format and Pairwise Formulation

Our proposed CoDA method accepts two input sequences $A \in \mathbb{R}^{\ell_a \times d}$ and $B \in \mathbb{R}^{\ell_b \times d}$, where $\ell_a, \ell_b$ are lengths of sequences $A$, $B$ repectively and $d$ is the dimensionaity of the input vectors, and returns pooled representations which are of equal dimensions. Note that CoDA is universal in the sense that it can be applied to both pairwise (cross) attention [Parikh et al., 2016, Seo et al., 2016] as well as single sequence attention. In the case of single sequence attention, $A$ and $B$ are often referred to the same sequence (i.e., self-attention [Vaswani et al., 2017]).

### 2.2 Dual Affinity Computation

We compute the pairwise affinity between each element in $A$ and $B$ via:

$$E_{ij} = \alpha \, F_E(a_i) F_E(b_j)^\top \tag{1}$$

which captures the pairwise similarity between each element in $A$ with each element of $B$. In this case, $F_E(.)$ is a parameterized function such as a linear/nonlinear projection. Moreover, $\alpha$ is a scaling constant and a non-negative hyperparameter which can be interpreted as a temperature setting that controls saturation. Next, as a measure of negative (dissimilarity), we compute:

$$N_{ij} = -\beta \, ||F_N(a_i) - F_N(b_j)||_{|\ell_1} \tag{2}$$

where $F_N(.)$ is a parameterized function, $\beta$ is a scaling constant, and $\ell_1$ is the $L1$ Norm. In practice, we may share parameters of $F_E(.)$ and $F_N(.)$. Note that $N_{ij} \in \mathbb{R}$ is a scalar value and the affinity matrix $N$ has equal dimension with the affinity matrix $E$. We hypothesize that capturing a flavour of dissimilarity (subtractive compositionality) is crucial in attentional models. The rationale for using the negative distance is for this negative affinity values to act as a form of gating (as elaborated subsequently).

## 2.3 Compositional De-Attention Matrix

In the typical case of vanilla attention, Softmax is applied onto the matrix $E \in \mathbb{R}^{\ell_A \times \ell_B}$ row-wise and column-wise, normalizing the matrix. Hence, multiplying the normalized matrix of $E$ with the original input sequence can be interpreted as a form of attentional pooling (learning to align), in which each element of $A$ pools all relevant information across all elements of $B$. For our case, we use the following equation:

$$M = tanh(E) \odot sigmoid(N) \tag{3}$$

where $M$ is the final (quasi)-attention matrix in our CoDA mechanism and $\odot$ is the element-wise multiplication between the two matrices.

**Centering of N.** Since $N$ is constructed by the negative L1 distance, it is clear that the range of $sigmoid(N) \in [0, 0.5]$. Hence, in order to ensure that $sigmoid(N)$ lies in $[0, 1]$, we center the matrix $N$ to have a zero mean:

$$N \to N - Mean(N) \tag{4}$$

Intuitively, by centering $N$ to zero mean, we are also able to ensure and maintain it's ability to both erase and retain values in $tanh(E)$ since $sigmoid(N)$ now saturates at 0 and 1, behaving more like a gate.

**Scaling of sigmoid(N).** A second variation, as an alternative to centering is to scale $sigmoid(N)$ by 2, ensuring that its range fall within $[0, 1]$ instead of $[0, 0.5]$.

$$M = tanh(E) \odot (2 * sigmoid(N)) \tag{5}$$

Empirically, we found that this approach works considerably well as well.

**Centering of E.** Additionally, there is no guarantee that $E$ contains values both positive and negative. Hence, in order to *ensure* that $tanh(E)$ is able to effectively express subtraction (negative values), we may normalize $E$ to have zero mean:

$$E \to E - Mean(E) \tag{6}$$

This normalization/centering can be interpreted as a form of inductive bias. Without it, we have no guarantee if the model converges to a solution where all values are only $E > 0$ or $E < 0$. Naturally, we also observe that the scalar value $\alpha$ in equation 1 acts like a temperature hyperparameter and when $\alpha$ is large, the values of $tanh(E)$ will saturate towards $\{-1, 1\}$.

**Intuition** Note that since the distance value takes a summation over vector elements, the value of $sigmoid(N) \in [0, 1]$ (centered) will saturate towards 0 or 1. Hence, this encodes a strong prior for either erasing (a.k.a '*de-attention*') or keeping the entire scores from $E$. Contrary to typical attention mechanism, $M$ is biased towards values $\{-1, 0, 1\}$ since $sigmoid(N)$ is biased towards $\{0, 1\}$ whilst $tanh(E)$ is biased towards $\{-1, 1\}$. Intuitively, $N$ (the negative affinity matrix) controls the deletion operation while $E$ controls whether we want to *add* or *subtract* a vector.

Additionally, we can consider that $sigmoid(N)$ acts as affinity gates. A higher dissimilarity (denoted by a large negative distance) will 'erase' the values on the main affinity matrix $tanh(E)$. The choice of other activation functions and their compositions will be discussed in Section **??**.

**Temperature** We introduced hyperparameters $\alpha, \beta$ in earlier sections that control the magnitude of $E$ and $N$. Intuitively, these hyperparameters control and influence the temperature of tanh and sigmoid functions. In other words, a high value of $\alpha, \beta$ will enforce a hard form of compositional pooling. For most cases, setting $\alpha = 1, \beta = 1$ may suffice. Note that the dimensionality of the vector contributes to the hard-ness of our method since this results in large values of $E, N$ matrices, this case, $\alpha, \beta$ may be set to $\leq 1$ to prevent $M$ from being too hard.

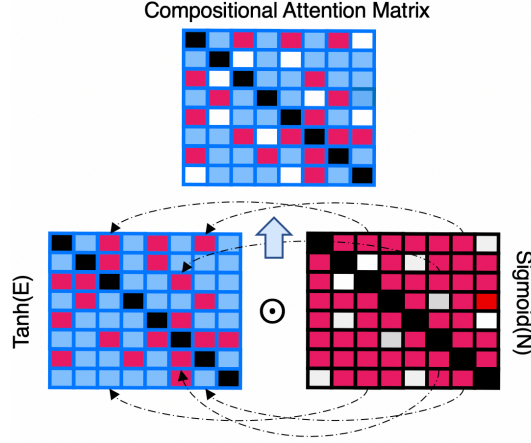

Figure 1: Illustration of our proposed Compositional De-Attention (CoDA) affinity matrix composition. Red represents positive values and blue represents negative values. White represents close to zero values.

## 2.4 Compositional Pooling

After which, we apply $M$ (the quasi-attention matrix) to input sequences $A$ and $B$.

$$A' = MB \ \ and \ \ B' = M^\top A \tag{7}$$

where $A' \in \mathbb{R}^{\ell_A \times d}$ and $B' \in \mathbb{R}^{\ell_B \times d}$ are the compositionally manipulated representations of $A$ and $B$ respectively. In the case of $A'$, each element $A_i$ in $A$ scans across the sequence $B$ and decides[2] whether to include/add $(+1)$, subtract $(-1)$ or delete $(\times 0)$ the tokens in $B$. Similar intuition is applied to $B'$ where each element in $B$ scans across the sequence $A$ and decides to add, subtract or delete the tokens in $A$. Intuitively, this allows for rich and expressive representations, unlike typical attentional pooling methods that softly perform weighted averages over a sequence.

## 2.5 Incorporating CoDA to Existing Models

In this section, we discuss various ways how CoDA may be used and incorporated into existing neural models.

**CoDA Cross-Attention**  Many models for pairwise sequence problems require a form of cross attention. In this case, CoDA is applied:

$$A', B' = CoDA(A, B) \tag{8}$$

where $A \in \mathbb{R}^{\ell_A \times d}$, $B \in \mathbb{R}^{\ell_B \times d}$ are two input sequences (e.g., document-query or premise-hypothesis pairs). $A' \in \mathbb{R}^{\ell_A \times d}$, $B' \in \mathbb{R}^{\ell_B \times d}$ are *compositionally* aligned representations of $A$ and $B$ respectively. Next, we use an alignment function,

$$F = [F(A', A); F(B', B)] \tag{9}$$

to learn cross-sentence feature representations. Note that $F(.)$ may be any parameterized function such as RNNs, MLPs or even simple pooling function. $[;]$ is the concatenation operator.

**CoDA Transformer**  Transformers [Vaswani et al., 2017] adopt self-attention mechanisms, which can be interpreted as cross-attention with respect to the same sequence. The original transformer[3]

equation of $A = softmax(\frac{QK^\top}{\sqrt{d_k}})V$ now becomes:

$$A = (tanh(\frac{QK^\top}{\sqrt{d_k}}) \odot sigmoid(\frac{G(Q,K)}{\sqrt{d_k}}))V \qquad (10)$$

where $G(.)$ is the negation of outer L1 distance between all rows of Q against all rows of K. We either apply centering to $(\frac{G(Q,K)}{\sqrt{d_k}}))V$ or $2 * sigmoid(\frac{G(Q,K)}{\sqrt{d_k}}))V$ to ensure the value is in $[0, 1]$. Finally, note that both affinity matrices are learned by transforming $Q, K, V$ only once.

## 3 Experiments

We perform experiments on a variety of NLP tasks including open domain question answering, retrieval/ranking, natural language inference, neural machine translation, sentiment analysis and text2code generation. This section provides experimental details such as experimental setups, results and detailed discussions.

### 3.1 Open Domain Question Answering

We evaluate CoDA on Open Domain QA. The task at hand is to predict an appropriate answer span in a collection of paragraphs. We use well-established benchmarks, SearchQA [Dunn et al., 2017] and Quasar-T [Dhingra et al., 2017]. Both dataset comprises of QA pairs with accompanying set of documents retrieved by search engines. For this experiment, we use the recently proposed and open source[4] DecaProp [Tay et al., 2018] as a base model and replace the context-query attention with our CoDA variation. We set the hyperparameters as closely to the original implementation as possible since the key here is to observe if CoDA enhanced adaptation can improve upon the original DecaProp. As competitors, we compare with the latest [Das et al., 2019], a sophisticated multi-step reasoner specially targeted at open domain QA, as well as the canonical $R^3$ model [Wang et al., 2017], AQA [Buck et al., 2017] and BiDAF [Seo et al., 2016].

| Model | Quasar-T | | SearchQA | |
|---|---|---|---|---|
| | EM | F1 | EM | F1 |
| GA | 26.4 | 26.4 | - | - |
| BiDAF [Seo et al., 2016] | 25.9 | 28.5 | 28.6 | 34.6 |
| AQA [Buck et al., 2017] | - | - | 38.7 | 45.6 |
| $R^3$ Reader-Ranker [Wang et al., 2017] | 34.2 | 40.9 | 49.0 | 55.3 |
| Multi-step-reasoner [Das et al., 2019] | 40.6 | 47.0 | 56.3 | 61.4 |
| DecaProp [Tay et al., 2018] | 38.6 | 46.9 | 56.8 | 63.6 |
| DecaProp + CoDA (Ours) | **41.3** | **49.7** | **57.2** | **63.9** |

Table 1: Experimental results on Open Domain Question Answering. DecaProp + CoDA achieves state-of-the-art performance on both datasets.

**Results** Table 1 reports the results on Open Domain QA. Most importantly, we find that CoDA is able to reasonably improve upon the base DecaProp model on the Quasar-T dataset ($+2.7\%$) while marginally improving performance on the SearchQA dataset. Notably, DecaProp + CoDA also exceeds specialized open domain QA models such as the recent Multi-step reasoner [Das et al., 2019] and achieves state-of-the-art performance on both datasets.

### 3.2 Retrieval and Ranking

We evaluate CoDA on a series of retrieval and ranking tasks. More concretely, we use well-established answer retrieval datasets (TrecQA [Wang et al., 2007] and WikiQA [Yang et al., 2015]) along with response selection dataset (Ubuntu dialogue corpus [Lowe et al., 2015]). These datasets are given a question-answer or message-response pair and are tasked to ranked the answer/responses to how likely they match the question.

For this experiment, we use a competitive baseline, DecompAtt [Parikh et al., 2016], as the base building block for our experiments. We train DecompAtt in pointwise model for ranking tasks with binary Softmax loss. We report MAP/MRR for TrecQA/WikiQA and top-1 accuracy for Ubuntu dialogue corpus (UDC). We train all models for 20 epochs, optimizing with Adam with learning rate 0.0003. Hidden dimensions are set to 200 following the original DecompAtt model. Batch size is set to 64.

|            | TrecQA     | WikiQA    | UDC  |
|------------|------------|-----------|------|
| D-ATT      | **80.6**/83.9 | 66.4/68.0 | 51.8 |
| D-ATT+CoDA | 80.0/**84.5** | **70.5/72.4** | **52.5** |

Table 2: Experimental results on Retrieval and Ranking.

**Results**   Table 2 reports our results on the retrieval and ranking task. D-ATT + CoDA outperforms vanilla D-ATT for most of the cases. On WikiQA, we observe a +4% gain on both MRR and MAP metrics. Performance gain on UDC and TrecQA (MRR) are marginal. Overall, the results on this task are quite promising.

## 3.3  Natural Language Inference

The task of Natural Language Inference (NLI) is concerned with determining whether two sentences entail or contradict each other. This task has commonly been associated with language understanding in general. We use four datasets, SNLI Bowman et al. [2015], MNLI [Williams et al., 2017], SciTail [Khot et al., 2018] and the newly released Dialogue NLI (DNLI) [Welleck et al., 2018]. Similar to the retrieval and ranking tasks, we use the DecompAtt model as the base model. We use an identical hyperparameter setting as the retrieval and ranking model but train all models for 50 epochs. We set the batch size to 32 for Scitail in lieu of a smaller dataset size.

| Model        | SNLI      | MNLI          | Scitail  | DNLI  |
|--------------|-----------|---------------|----------|-------|
| D-ATT        | 84.61     | 71.34/71.97   | 82.0     | 88.2  |
| D-ATT + CoDA | **85.71** | **72.19/72.45** | **83.6** | **88.8** |

Table 3: Experimental results of accuracy on Natural Language Inference.

**Results**   Table 3 reports the results of our experiments on all four NLI tasks. Concretely, our results show that CoDA helps the base DecompAtt on both datasets. Notably, DecompAtt + CoDA outperforms the state-of-the-art result of 88.2% on the original DNLI dataset leaderboard in [Welleck et al., 2018].

## 3.4  Machine Translation

We evaluate CoDA-Transformer against vanilla Transformer on Machine Translation (MT) task. In our experiments, we use the IWSLT'15 English-Vietnamese dataset. We implement CoDA-Transformer in Tensor2Tensor[5]. We use the `transformer_base_single_gpu` setting and run the model on a single TitanX GPU for 50K steps and using the default checkpoint averaging script. Competitors include Stanford Statistical MT [Luong and Manning], traditional Seq2Seq + Attention [Bahdanau et al., 2014], and Neural Phrase-based MT [Huang et al., 2017].

| Model                     | BLEU      |
|---------------------------|-----------|
| Luong & Manning (2015)    | 23.30     |
| Seq2Seq Attention         | 26.10     |
| Neural Phrase-based MT     | 27.69     |
| Neural Phrase-based MT + LM | 28.07   |
| Transformer               | 28.43     |
| CoDA Transformer          | **29.84** |

Table 4: Experimental results on Machine Translation task using IWSLT'15 English-Vietnamese dataset.

**Results** Table 4 reports the result on our MT experiments. We observe that CoDA improves the base Transformer model by about $+1.4\%$ BLEU points on this dataset. Notably, CoDA Transformer also outperforms all other prior work on this dataset by a reasonable margin.

## 3.5 Sentiment Analysis

We compare CoDA-Transformer and Vanilla Transformer on word-level sentiment analysis. We use the IMDb and sentiment tree-bank (SST) sentiment dataset. We implement CoDA-Transformer in Tensor2Tensor and compare using the *tiny* default hyperparameter setting for both models. We train both models with 2000 steps.

| Model | IMDb | SST |
|---|---|---|
| Transformer | 82.6 | 78.9 |
| CoDA Transformer | **83.3** | **80.6** |

Table 5: Experimental results on IMDb and SST Sentiment Analysis.

**Results** We observe that CoDA-Transformer outperforms vanilla Transformer on both datasets. Note that this implementation uses Byte-pair Encoding/No pretrained vectors and therefore is not comparable with all other works in literature that use this IMDb and SST datasets.

## 3.6 Mathematical Language Understanding (MLU)

We evaluate CoDA on the arithmetic MLU dataset [Wangperawong, 2018], a character-level transduction task[6]. The key idea is to test the compositional reasoning capabilities of the proposed techniques. Example input to the model is $x = 85, y = -523, x * y$ and the corresponding output is $-44455$. A series of variations are introduced, such as permutation of variables or introduction of other operators. The dataset comprises of 12 million of these input-output pairs.

**Implementation** Following [Wangperawong, 2018], we trained a CoDA Transformer model on this dataset for $100K$ steps. Evaluation is performed during accuracy per sequence which assigns a positive class when there is an exact match.

| Model | Acc |
|---|---|
| Transformer[†] | 76.1 |
| Universal Transformer[†] | 78.8 |
| CoDA Transformer | **84.3** |

Table 6: Experimental results on Mathematical Language Understanding (MLU). † denotes results reported from [Wangperawong, 2018].

**Results** Table 6 reports the results of our experiments. We observe that our CoDA Transformer achieves a sharp performance gain (+8.2%) over the base Transformer model. Moreover, the CoDA Transformer achieves almost full accuracy (solving this task) on this dataset, showing the advantages that CoDA has on compositional reasoning.

## 3.7 Program Search / Text2Code Generation

We report additional experiments on language to code generation. We use the AlgoLisp dataset from [Polosukhin and Skidanov, 2018], which is implementation of the problems in a Lisp-inspired programming language. Each problem has 10 tests, where each test is input to be fed into the synthesized program and the program should produce the expected output. We frame the problem as a sequence transduction task.

Similar to other experiments, our implementation is based on the Tensor2Tensor framework. We train Transformer and CoDA Transformer for $100K$ steps using the *tiny* setting. The evaluation metric is accuracy per sequence which means the model only gets it correct if it generates

the entire sequence correctly. Baselines are reported from [Polosukhin and Skidanov, 2018].

| Model | Acc |
|---|---|
| Attention Seq2Seq | 54.4 |
| Seq2Tree + Search | 86.1 |
| Transformer | 96.8 |
| CoDA Transformer | **97.7** |

Table 7: Experimental results on Text2Code Generation task (AlgoLisp).

**Results** Table 7 reports results on the Text2Code task. Our CoDA Transformer outperforms the base Transformer by about $+0.9\%$ and overall achieving state-of-the-art results on this task. Similar to the MLU task, CoDA Transformer comes close to solving this problem.

## 4 Analysis

**Centering of $E$ and $N$.** We further examine the need for centering of $E$ and $N$ and show the experimental results on NMT, WikiQA, SciTail and DNLI datasets in table 8. We observe that centering $E$ and $N$ does not help in performance in most of the cases, except for the DNLI dataset (but the gap is mimimal though). We conclude that instead of forcing the balance between the positive and negative values in $E$ and $N$, it is better for the model to learn when to add (positive value), negate (negative value) or ignore (zero value) the tokens.

| Function | NMT | WikiQA | SciTail | DNLI |
|---|---|---|---|---|
| $F_{Tanh}(E) \odot F_{Sigmoid}(N)$ | **27.8** | **70.4 / 71.2** | **85.3** | 86.5 |
| $F_{Tanh}(E') \odot F_{Sigmoid}(N)$ | 26.9 | 68.1 / 68.3 | 84.5 | **86.8** |
| $F_{Tanh}(E') \odot F_{Sigmoid}(N')$ | 27.3 | 69.3 / 70.2 | 85.0 | 86.7 |

Table 8: Ablation study (development scores) of centering $E$ and $N$. $E'$ and $N'$ refer to centered/mean zeroed values of $E$ and $N$. Model architectures remain identical to experiment section.

**Ablation study** Table 9 shows the development scores of different compositions on Machine Translation task using IWSLT'15 dataset. We observe that the composition of applying $Tanh$ on $E$ and $Sigmoid$ on $N$ helps our proposed attention mechanism achieve the best performance compared to other compositions.

| Function | BLEU |
|---|---|
| $F_{Tanh}(E) \odot F_{Sigmoid}(N)$ | **27.8** |
| $F_{Tanh}(E) \odot F_{Tanh}(N)$ | 23.1 |
| $F_{Tanh}(E) \odot F_{Arctan}(N)$ | 21.9 |
| $F_{Tanh}(E) \odot F_{Algebraic}(N)$ | 26.1 |
| $F_{Sigmoid}(E) \odot F_{Tanh}(N)$ | 21.9 |
| $F_{Sigmoid}(E) \odot F_{Sigmoid}(N)$ | 27.5 |
| $F_{Sigmoid}(E) \odot F_{Arctan}(N)$ | 25.1 |
| $F_{Sigmoid}(E) \odot F_{Algebraic}(N)$ | 26.3 |

Table 9: Ablation study (development scores) of various composition functions on MT task. Model architectures remain identical to experiment section.

## 5 Visualization

In order to provide an in-depth study of the behaviour our proposed mechanism, this section presents a visual study of the CoDA mechanism. More concretely, we trained a model and extracted the matrices $tanh(E)$, $sigmoid(N)$ and $M$. Figure 2 illustrates some of these visualizations.

We make several key observations. First, the behaviour of $sigmoid(N)$ is aligned with our intuition, saturating at $\{0, 1\}$ and acting as gates. Second, the behaviour of $tanh(E)$ concentrates around 0 but spreads across both negative and positive values. Lastly, the shape matrix $M$ follows $tanh(E)$ quite closely although there are a larger percentage of values close to 0 due to composing with $sigmoid(\text{N})$.

At convergence, the model learns values of $tanh(E)$ which are close to 0 and more biased towards negative values. This is surprising since we also found that $tanh(E)$ are saturated, i.e., $\{-1, -1\}$ at the early epochs. We found that the model learns to shrink representations, allowing $tanh(E)$ to

have values closer to 0. Finally we note that the shape of $M$ remains similar to $tanh(E)$. However, the distribution near 0 values change slightly, likely to be influenced by the $sigmoid(N)$ values.

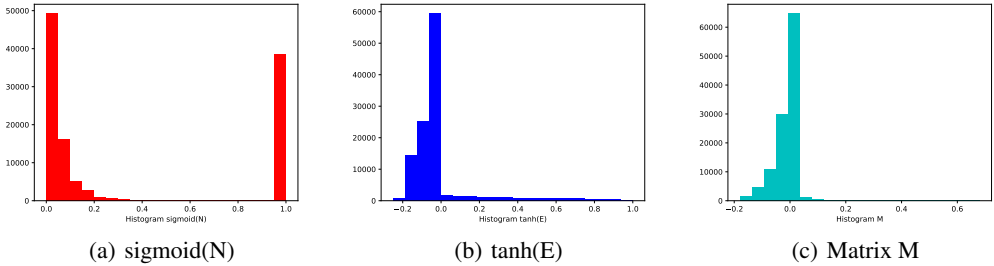

| (a) sigmoid(N) | (b) tanh(E) | (c) Matrix M |

Figure 2: Visualization at $d = 200$.

## 6   Related Work

Attention [Bahdanau et al., 2014] is a well-established building block in deep learning research today. A wide spectrum of variations have been proposed across the recent years, including Content-based [Graves et al., 2014], Additive [Bahdanau et al., 2014], Location-based [Luong et al., 2015], Dot-Product [Luong et al., 2015] and Scaled Dot-Product [Vaswani et al., 2017]. Many of these adaptations vary the scoring function which computes alignment scores. Ultimately, Softmax operator normalizes the sequence and computes relative importance. In essence, the motivation of attention is literally derived from its naming, i.e., to *pay attention* to certain parts of the input representation.

In very recent years, more sub-branches of attention mechanisms have also started to show great promise across many application domains. Self-attention [Xu et al., 2015, Vaswani et al., 2017] has been shown to be an effective replacement for recurrence/convolution. On the other hand, Bidirectional attention flow [Seo et al., 2016] is known to be effective at learning query-document representations. Decomposable attention [Parikh et al., 2016] provides a strong inductive prior for learning alignment in natural language inference. A common denominator of these recent, advanced attention mechanisms is the computation of an affinity matrix which can be interpreted as a fully connected graph that connects all nodes/tokens in each sequence.

The extent of paying attention is also an interesting area of research. An extreme focus, commonly referred to as hard attention [Xu et al., 2015] tries to learn discriminate representations that focus solely on certain targets. Conversely, soft attention [Bahdanau et al., 2014] access and pools across the entire input. There are also active research pertaining to the activation functions of attention mechanisms, e.g., Sparsemax [Martins and Astudillo, 2016], Sparsegen [Laha et al., 2018] or EntMax [Peters et al., 2019]. However, influencing the sparsity of softmax can be considered an orthogonal direction from this work.

All in all, the idea of attention is to learn relative representations. To the best of our knowledge, there have been no work to consider learning attentive representations that enable negative representations (subtracting) during pooling. Moreover, there is also no work that considers a dual affinity scheme, i.e., considering both positive and negative affinity when learning to attend.

## 7   Conclusion

We proposed a new quasi-attention method, the compositional de-attention (CoDA) mechanism. We apply CoDA across an extensive number of NLP tasks. Results demonstrate promising results and the CoDA-variations of several existing state-of-the-art models achieve new state-of-the-art performances in several datasets.

## Footnotes

[2]We emphasize that this is still done in a soft manner.

[3]We find that, for some tasks, removing the scaling factor of $\frac{1}{\sqrt{d_k}}$ works better for our CoDA mechanism.

[4] `https://github.com/vanzytay/NIPS2018_DECAPROP`.

[5]`https://github.com/tensorflow/tensor2tensor`.

[6]This task is readily available on the Tensor2Tensor framework.

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
