[Reviews · NeurIPS 2019]

Reviewer 1



UPDATE after reading author rebuttal: I am looking forward to the more comprehensive evaluation that you are carrying out. Regarding Q3, please include details of the setup in the main paper. Also, more analysis needed regarding why zeroes are predominant in M in the main paper (also a point raised by R3) - rather than speculation or hypothesis. Overall, my opinion of the paper does not change and feel it is a good direction of research. Detailed comments: 1. This paper proposes an alternative to softmax-based attention mechanism - a quasi-attention technique : A dual affinity matrix approach is proposed compared to the usual single affinity matrix. One affinity matrix is created from the pairwise similarity computation. The second one is computed based on l1-distance based dissimilarity (with a negative). Application of sigmoid on the second ensures either deletion or propagation. tanh operation on the first leads to a choice between addition or subtraction. Both operation together help choose between addition, subtraction or deletion. These operations can be composed to be incorporated in the cross-attention and self-attention mechanism in transformers. With this kind of composition, the expressiveness of models in increased. Minor comment : This is not the first work on softmax-less attention. There were works earlier like sparsemax [Martins et al 2016], sparsegen [Laha et al 2018], which incorporate weights which are positive or zero. However, this work first proposes the idea of the inclusion of negative (that is subtraction), hence, is novel in that way. 2. Experimentation and evaluation in six tasks through multiple frameworks (cross-attention and self-attention): The experiments are reasonable and includes a wide range of tasks. However, there are few things that would have made it comprehensive: a) Experimentation on SNLI dataset for Natural Language Inference which is more well-known and considered as a benchmark for NLI tasks. b) Experimentation on EN-FR, EN-DE, FR-EN, DE-EN, etc. are much needed for better comparison in NMT as these are more common languages. c) On Sentiment analysis, SST dataset will make the analysis complete. 3. Comments regarding analysis and visualization: Even though there is good mathematical intuition regarding the choice of the affinity matrices and the composition functions, the visualization is rather surprising that the Matrix M values tend to be close to 0. a) How the visualization is constructed is not clear. Are the values considered for a particular trained model for all data points? b) More insights and analysis is needed as to why deletion is more prevalent compared to addition or subtraction (M has more zero values). Will having higher alpha/beta values help? No mention of these hyperparameters in the experimental setting. Only mention is in the main section where they are both set to 1. It is also not clear why you would not want M to have harder extreme values? c) The form of Eq 9. that was used for experiments is not mentioned (or possibly missed by the reviewer). 4. Minor Typos: a) dimensionaity --> dimensionality. b) Line 117 missing reference. References: [Martins et al 2016] Martins and Ramón F. Astudillo. From softmax to sparsemax: A sparse model of attention and multi-label classification. [Laha et al 2018] Anirban Laha, Saneem Ahmed Chemmengath, Priyanka Agrawal, Mitesh Khapra, Karthik Sankaranarayanan, and Harish G Ramaswamy. On Controllable Sparse Alternatives to Softmax. Assessment (out of 10): Originality: 8 Quality: 7 Clarity: 9 Significance: 8

Reviewer 2



Updated review in light of author feedback: The authors have recognized that their terminology was unclear, that the experiments were missing details, and that there could be additional experiments from T2T. They have indicated that they plan to address all of these concerns in the camera-ready version. Assuming they do so, I assume the paper will be improved enough to warrant an acceptance, so I am raising my score to 7 which leads to an "accept" consensus among reviewers. Summary: This paper proposes an alternative form of attention which replaces the standard "query-key dot-product followed by softmax" form of attention which is widely used. Given two sequences A and B, CoDA an affinity matrix A and a dissimilarity matrix N and uses them to compute M = tanh(A)*sigmoid(N) to form a gated subtract, delete, or add operation. M is then used as usual for attention, without the use of softmax. The method is tested by replacing standard attention on many networks and on many datasets, and is shown to consistently outperform standard attention. Review: Overall it is an interesting and worthwhile pursuit to consider alternatives to the ubiquitous form of attention. CoDA presents some interesting possibilities due to the fact that it avoids softmax and instead allows for the possibility of adding, subtracting, or gating out values from each sequence. I also appreciate the experimental approach which replaces something standard with an alternative and shows that it consistently helps. For these reasons, I think this is a good submission. On the negative side, as I expand on below, the notation is unclear and oftentimes self-contradictory. It definitely would need to be cleaned up before acceptance. Further, there are various alternatives proposed for the exact form of CoDA - whether to subtract the mean before applying the sigmoid or multiply by 2; whether to scale before applying self-attention; various alternatives for the function "F" for standard attention, etc. However, no details are given as to which were used when and no comparison is given to the performance of each alternative. If different versions of CoDA were used on different datasets, that is worrisome becuase it implies limited generality of the approach. Similarly, most of the results are using the tensort2tensor codebase but a specific set of datasets from tensor2tensor were chosen. Why these? Did it not work on the other datasets? I also think it would be very helpful to see a comparitive visualization, for some illustrative examples, of standard attention and the CoDA attention matrix "M". This could go in the supplementary materials. I will give this paper a weak accept and if the above improvements/clarifications are made I will be happy to raise my review to an accept. Specific comments: - "Softmax is applied onto the matrix E row-wise and column-wise, normalizing the matrix." This is not true, it is only applies row-wise or column-wise (depending on whether you are "pooling" the elements of A or B). For example, if you intend to pool the elements of B, you'd multiply E*B and normalize the rows of E with softmax, not the columns. - When you center N (or E) to have zero mean, are you computing the average across the entire matrix N? Or on a per-row or per-column basis? This would be more clear if you replaced "Mean" in (4) with an explicit nested summation. - "in order to ensure that tanh(E) is able to effectively express subtraction" This statement is confusing without context because tanh(E) has not been used to perform any kind of subtraction yet. - I think there are some issues with the notation in 2.4. In (7), A' and B' are computed using B and A respectively, but the text says "A' and B' are the compositionally manipulated representations of A and B respectively." Also, the text reads "In the case of A' each element A_i in A scans across the sequence B and decides whether to include/add (+1), subtract (−1) or delete (×0) the tokens in B." Don't you mean "...each row M_i in M scans..."? A does not appear anywhere in the expression for A' in (7). - You never explicitly define what the CoDA function does (used in (8)). I assume you mean you compute M via (3) and then A' and B' via (7), but this should be made explicit. - You introduce an "alignment function" F without describing its purpose. Please expand. - You write "We either apply centering to ... or 2*...". I think you mean "We either apply centering to ... or compute 2*...". It sounds like you are applying centering to 2*... which I don't think is what you mean. - In some cases, the results tables are presented without including all prior work which makes statements like "CoDA Transformer also outperforms all other prior work on this dataset by a reasonable margin." For example, for IWSLT'15 EnVi, both https://arxiv.org/abs/1809.08370 and https://www.aclweb.org/anthology/N19-1192 obtained BLEU scores of 29.6, which is not far from the 29.84 obtained in this paper. To me, the most important thing is that CoDA improves over regular attention as a simple swap-in replacement, so I think the results should emphasize this -- not marginal SoTA. - For visualization, it would be more informative to show a few examples of what M is on a few representative sequences. - In the related work, you cite Xu et al. 2015 as a reference for "self-attention". I don't believe there is any self-attention in Xu et al. - Given that you are using tensor2tensor in some experiments, why not include results on other standard datasets like WMT translation etc? - In which cases do you use centering vs. multiplying by 2 ((4) or (5))? Footnote 2 reads that "removing the scaling factor [sometimes] works better". When did you and didn't you use the scaling factor? What function/network do you use for F?

Reviewer 3



This paper introduces Compositional De-Attention (CoDA), a novel attention mechanism that learns to add, subtract or nullify input vectors instead of simply make a weighted sum of them. They do so by multiplying the output of a tanh and a sigmoid function resulting in a attention mechanism biased towards -1, 0 and 1 instead of the softmax’s bias towards 0 and 1. The authors qualified CoDA as quasi-attention since it does not make use of the usually used softmax function. They demonstrate through extensive experiments the power of this novel attention mechanism and gain state of the art on several datasets. Originality: The idea of mixing the sigmoid function and a tanh function in order to capture negative, zero and positive correlations in attention mechanisms is quite novel. None of the existing works on attention mechanisms exploit this idea. The authors clearly differentiate their work from previous works. Related work has been cited adequately. Quality: Experimental results show that CoDA gives an edge compared with traditional attention mechanisms. Indeed, they evaluate it on several tasks and datasets as enumerated in the Contributions section and they indirectly gain state of the art on many of these datasets by doing so. While they evaluate on a wide variety of tasks, it seems to be some kind of work in progress since only a few datasets per task are evaluated. For instance, for the task of open domain QA, they do not evaluate on the SQuAD dataset which one dataset usually used for this task. By evaluating on this dataset, they could also have compared with BERT (Transformer) architecture which they do not on this task. In Section 4, we can see through figures that the attention learned gravitates towards 0 and small negative values and is quite similar to what the tanh function alone outputs. This raises the question whether the sigmoid has any impact on the final outcome. An ablation study should be done on that. Finally, visualization of the learned attention weights on examples could have been nice for interpretability. Clarity: The paper was well organized. Sections follow the usual order of NIPS papers. The proposed approach is well described. The authors did not correctly fill the reproducibility form by answering No to all questions. Few aesthetic comments: Use \tanh, \text{sigmoid} and \text{softmax}. Equation 2 : The usual notation for the L1-norm is a subscript 1 instead of ${|\ell_1}$. Equation 10 and Line 146 : Use \left( and \right). Line 117 : Undefined section label. Line 237 : Did you mean {-1, 1} ? Significance: Notwithstanding my previous comments, the results obtained by the authors are really promising. I am sure this paper will motivate other researchers to follow their path and use other activation functions in attention mechanism. The fact that only changing the activation function can improve drastically the results also might create a subfield researching the effect of activation functions in attention mechanisms. Post rebuttal response ----------------------------- The author responded well to my concerns.

[Author Response · NeurIPS 2019]

**To All Reviewers**   We wholeheartedly thank all reviewers for all valuable and insightful feedback, along with taking the time to review our submission. We have provided detailed comments to each reviewer below.

**To Reviewer 1**   Thanks for the valuable and insightful feedback!

Regarding the suggestion about more experiments on more datasets, we full wholeheartedly agree that more experiments will definitely improve the comprehensiveness of the paper. We are running experiments on these tasks right now and include results on these datasets (SST, SNLI, WMT En-De etc.) by the camera ready version as supplementary material.

Pertaining to Q3, weights are extracted from a trained model (on NLI) on sampled $1K$ datapoints from the dev set. Similar graphs appear with repeated sampling. Regarding deletion being prevalent, our hypothesis is that only tokens that are particularly significant will be compositionally (add/subtracted). For most words, the sigmoid function provides flexibility of deleting tokens.

Regarding the value of $\alpha, \beta$ in equation 1 and equation 2, we set them to 1 in the experiments. The higher value of $\alpha, \beta$ is, the 'harder' the compositional pooling becomes. The hardness of CoDA required is possibly related and analogous to hard vs soft attention and can be domain dependent. For most language tasks, we find that not biasing CoDA towards being hard is quite sufficient. Not all tokens are important, so CoDA maintains the flexibility of standard attention while enabling arithmetic compositionality. We will include more supplementary distribution visualisations on different tasks in the revised version.

Regarding the form of Equation 9 used in the experiments, we used one layer non-linear projection network to compute the pairwise affinity. We thank the reviewer for pointing it out and we will mention it clearly in the revision.

For other comments such as references or typos, we will correct and add them in the revision. We will also be sure to include a discussion about sparsemax and softgen.

**To Reviewer 2**   Thanks for the insightful and valuable feedback!

Regarding the evaluation on Tensor2Tensor, IWSLT En-Vi was chosen because of the size and resource limitations. Please be assured that the tasks were not cherry-picked and we have not experienced any failure cases on T2T tasks yet. We also had success on the arithmetic T2T and subject-verb agreement tasks but did not report due to lack of space. We will prepare detailed supplementary material to cater to extra experiments. Moreover, we are currently running WMT En-De and WMT En-Ro. This may take sometime due to our limited hardware but will definitely be ready by the camera-ready version.

Regarding the form of CoDA, we use the following version our experiments, i.e. subtract the mean before applying the sigmoid or tanh, and they are scaled before applying self-attention. We will make this absolutely clear in the final version. We left several variations open in our technical exposition since certain hyperparameters (e.g., $\alpha$) could allow practitioners to control properties (i.e., hardness) of CoDA. We apologize for any confusion. In the revision, we will also include more ablation studies of different alternations of CoDA form, providing better guidance for usage of CoDA configurations. We will also include supplementary visualisation as requested.

Regarding the confusion of using notions and references in the papers, as pointed out in the detailed comments, we thank the reviewer for pointing them out. We will correct and clean them in the revision.

**To Reviewer 3**   Thanks for the insightful and valuable feedback!

Our intuition is that tanh saturates at $\{-1, 1\}$, which doesn't allow the model the flexibility to *delete* (erase) tokens. Sigmoid provides this flexibility to our model. Some early ablation results on retrieval tasks are reported in Table 1. We will include more comprehensive ablations in the final version.

Thanks for the suggestion about adding a visualization of the learned attention weights on examples for interpretability. We will definitely include it in the revision. Pertaining aesthetic comments, thanks for pointing them out and we will be sure to correct them in the revision.

| Method | TrecQA | WikiQA |
|---|---|---|
| *tanh* only | 66.78 / 73.49 | 67.13 / 67.81 |
| CoDA | 79.84 / 84.78 | 68.07 / 68.28 |

Table 1: Dev set results of *tanh* only versus CoDA (*tanh * sigmoid*).

[Meta-Review · NeurIPS 2019]

The present paper proposes a variant of an attention mechanism (called a quasi-attention mechanism) that allows in a soft manner to add (+1), subtract (-1) or erase (×0) information from the attended input. This new feature gives to the mechanism the capability of learning negative correlations in addition to the usual positive and zero correlations. The definitions are mathematically sounded and the authors give a nice explanation of the underlying intuition. Experimentations on various tasks illustrate the behaviour of the mechanism and show that it also improves the model performance, even beating the state of the art on some datasets. This paper is a clear accept according to NeurIPS standards, however after discussion with the reviewer, I will not recommend it for a talk since it is basically a nice improvement in an already fully explored direction.